# Exploring the association between precipitation and population cases of ocular toxoplasmosis in Colombia

**Laura Boada-Robayo[1], Danna Lesley Cruz-Reyes[2], Carlos Cifuentes-González[1], William Rojas-Carabali[1], Ángela Paola Vargas-Largo[1], Alejandra de-la-Torre[1] ***

**1** Universidad del Rosario, School of Medicine and Health Sciences, Neuroscience (NEUROS) Research Group, Bogotá, Colombia, **2** Universidad del Rosario, School of Medicine and Health Sciences, Clinical Research Group, Bogotá, Colombia

* alejadelatorre@yahoo.com

**Data Availability Statement:** All relevant data are within the manuscript, Supporting information files and in the repositorium of the Universidad del

## Abstract

### Background

Previous studies suggest a relationship between precipitation and ocular toxoplasmosis (OT) reactivation and congenital toxoplasmosis infection. We aimed to investigate the relationship between precipitation and the frequency of new OT cases in Colombia from 2015 to 2019.

### Methodology

This retrospective cohort study analyzed data obtained from a claims-based database created by the Colombian Ministry of Health and national registries of precipitation of the Institute of Hydrology, Meteorology, and Environmental Studies. We estimated the daily number of OT cases, interpolating data from the average number of annual cases from 2015 to 2019. Then, we compared exposures (mean daily precipitation) in the case period in which the events (interpolated OT new cases) occurred by a quasi-Poisson regression, combined with a distributed lag non-linear model to estimate the non-linear and lag–response curve.

### Principal findings

In the 5-year analysis, there were 1,741 new OT cases. Most of the cases occurred in 2019, followed by 2015 and 2018. New OT cases among departments were significantly different (*P*< 0.01). The cumulative exposure-response curve was decreasing for most departments. Nevertheless, in Chocó, Bogotá, Cesar, Cauca, and Guajira, when a certain amount of precipitation accumulates, the relative risk (RR) increases, which was contrary to the pattern observed in the other regions. The response curves to the one-day lag showed that precipitation influences the RR; however, the trends vary by department. Finally, an increasing trend in the number of cases was directly proportional to precipitation in Guajira, Atlántico, Norte de Santander, Santander, Caquetá and Quindío (r = 0.84; *P*< 0.05).

Rosario https://doi.org/10.34848/DLTIAB; https://doi.org/10.34848/U13ZND; https://doi.org/10.34848/UCNLLM.

**Funding:** The proofreading was supported by Universidad del Rosario. The funders had no role in study design, data collection and analysis, decision to publish, or preparation of the manuscript.

**Competing interests:** The authors have declared that no competing interests exist.

## Conclusions

Precipitation influenced the RR for new OT cases. However, varying trends among geographical regions (departments) lead us to hypothesize that other sociodemographic, behavioral, and environmental variables, such as wind and water contamination, could influence the RR.

## Author summary

We analyzed data obtained from the Colombian Ministry of Health and the national meteorology center to determine the effect of precipitation on the new cases of ocular toxoplasmosis (OT). The number of interpolated daily cases was estimated and compared with the exposure (precipitation) by a quasi-Poisson regression, combined with a distributed lag non-linear model to estimate the lag, non-linear response curve, and a Pearson correlation test. In the 5-year study period, 1,741 new cases of OT were reported. We found significant differences in the trends among all departments, with most departments showing decreasing cumulative exposure-response curves. However, in Chocó, Bogotá, Cesar, Cauca, and Guajira, when a certain amount of precipitation accumulates, the relative risk (RR) increases, contrary to the pattern observed in the rest of the country. The response curves at one-day lag showed that precipitation influences the RR; however, trends vary by department. We found a positive correlation between the number of cases and precipitation ($\beta = 0.03$; $P < 0.05$), indicating that precipitation affects the RR of new cases of OT. However, inconsistent trends exist across geographical regions, leading us to hypothesize that other sociodemographic, behavioral, and environmental variables, such as wind and water contamination, might influence the RR.

## Introduction

*Toxoplasma gondii (Tg)*, an intracellular protozoan belonging to the Apicomplexa family, is responsible for toxoplasmosis [1]. The life cycle of *Tg* allows it to survive in humid conditions. Warm-blooded animals, including humans, are intermediate hosts and felines are the definitive hosts of *Tg*. Unsporulated oocysts are present in the cats' feces. Oocysts take 1–5 days to sporulate and become infective in the environment. Intermediate hosts become infected after ingesting soil, water, or plant materials contaminated with oocysts. Inside the host's gastrointestinal tract, oocysts become tachyzoites, which then migrate to the neural and muscle tissues and develop into tissue cyst bradyzoites. Cats become infected after consuming intermediate hosts harboring tissue cysts or sporulated oocysts. Humans become infected by eating undercooked meat of animals with tissue cysts, consuming food or water contaminated with oocysts, blood transfusion, organ transplantation, and transplacentally from mother to fetus. The parasites form tissue cysts in the human host, most commonly in the skeletal muscle, myocardium, brain, and retina [2]. In humans, the infection can be congenital or postnatally acquired [3]. The clinical spectrum varies from asymptomatic to local manifestations, such as ocular toxoplasmosis (OT) or systemic disease [4].

Although OT is one of the leading causes of infectious posterior uveitis worldwide [5], it is considered a neglected tropical disease. In South America, retinochoroiditis reactivation episodes are more frequent and severe than those in Europe and North America due to the

virulent serotypes in our region [6]. The annual incidence of OT was calculated previously as three new episodes per 100,000 inhabitants in Colombia [7]. Furthermore, *Tg* is the leading infectious cause of visual impairment in immunocompetent patients [8] and Colombia's second most common cause of congenital blindness [9].

A previous study has shown that the mean annual rainfall significantly correlates with the incidence of congenital toxoplasmosis [10]. Furthermore, another study found a positive correlation between the annual precipitation and toxoplasmic retinochoroiditis reactivation rate [11]. This correlation could exist because oocyst survival is higher in humid, warm soil than under dry conditions [12]. Quantitative estimation of the viability of *Tg* oocysts in soil demonstrated that, after 100 days, 7.4% and 43.7% of oocysts survived under dry and damp conditions, respectively [12]. *Tg* oocysts survive up to 54 months in cold water [13].

The literature describing the effect of precipitation and risk of OT is limited; however, it is hypothesized that higher rainfall is associated with increased rates of congenital toxoplasmosis and OT recurrences [10,11]. Therefore, considering that Colombia is one of the rainiest countries worldwide [14], this study aimed to investigate the relationship between precipitation and new cases of OT from 2015 to 2019 in our country. We hypothesize that higher precipitation is a risk factor for developing OT.

## Methods

### Ethics statement

This study adheres to the ethical principles for human research established by the Helsinki Declaration, Belmont Report, and Colombian Resolution 008430 of 1993. According to the risks contemplated in resolution 8430 from 1993, this investigation is considered without risks. The information in the databases used in this article is freely accessible and is available for research purposes. In the same way, their coding system ensures data confidentiality.

### Design

In this retrospective cohort study, we analyzed the correlation of the daily mean of new OT cases among Colombian patients with the daily average precipitation in Colombia between 2015 and 2019.

### Population data

Colombia has a tropical forest and a tropical monsoon climate based on the Köppen–Geiger classification system; thus, it has a huge climatic diversity [15]. Colombia is administratively and politically divided into 32 geographical regions with different population densities that share cultural and economic characteristics, which are called "departments." During the study's observation period, Colombia had between 46,313,898 and 49,395,678 inhabitants. The distribution in each of the departments is shown in Table 1.

We obtained the study data from the national database created by the Colombian Ministry of Health, which is known as the System of Information of Social Protection (SISPRO) [16]. Its function is to store, process, and systematize Colombian citizens' information to make decisions that support the development of effective policies and monitoring in sectors, such as health, pensions, occupational risks, and social promotion. Health data are collected and codified by medical staff during each medical contact (inpatient or outpatient) from private and public health providers and insurers using the International Statistical Classification of Diseases 10th Revision (ICD-10). The Individual Records of Health Service Provision (RIPS) groups all demographic and clinical data [17]. According to a recent report, the Colombian

**Table 1. Distribution of precipitation, population, and ocular toxoplasmosis cases by department.**

| Department | Precipitation[a] | Population[b] | OT cases[c] | OT cases per inhabitants |
|---|---|---|---|---|
| | Daily mean in mm (SD) T = Total precipitation in 5 years in mm. | Mean 5 years [range] | Annual mean (SD) | Average cases in 5 years per million inhabitants |
| Amazonas | nd | 75,144.2 [72,485–77,753] | nd | nd |
| Antioquia | 8.02 (16.5) T = 104,436 | 6'320,083.2 [6'134,953–6'550,206] | 88.2 (19.0) | 13.96 |
| Arauca | nd | 255,352.6 [239,772–280,109] | 3.40 (3.85) | 13.31 |
| Atlántico | 2.67 (9.94) T = 7,141 | 2'492,540 [2'393,557–2'638,151] | 6.40 (5.22) | 2.57 |
| Bogotá, DC. | 2.39 (5.28) T = 26,641 | 7'383,413.8 [7'273,265–7'592,871] | 106 (38.5) | 14.36 |
| Bolívar | 7.23 (18.2) T = 21,462 | 2'048,620.6 [1'993,760–2'130,512] | 9.20 (6.22) | 4.49 |
| Boyacá | 3.21 (7.65) T = 128,436 | 1'209,827 [1'193,206–1'230,910] | 8.80 (6.22) | 7.27 |
| Caldas | 9.78 (19.5) T = 15,049 | 993,907.2 [984,360–1'008,344] | 39.2 (15.6) | 39.44 |
| Caquetá | nd | 401,559.6 [398,725–406,142] | 22.2 (15.9) | 55.28 |
| Casanare | nd | 412,039.2 [396,320–428,563] | 3.40 (1.14) | 8.25 |
| Cauca | 5.40 (8.71) T = 13,723 | 1'449,288 [1'420,313–1'478,407] | 20.6 (7.13) | 14.21 |
| Cesar | 4.65 (13.4) T = 49,399 | 1'173,191 [1'114,269–1'252,398] | 23.4 (17.2) | 19.95 |
| Chocó | 12.8 (23.8) T = 44,973 | 525,824.2 [509,240–539,933] | 4.40 (2.97) | 8.37 |
| Córdoba | 4.00 (12.0) T = 88,767 | 1'765,363.2 [1'726,287–1'808,439] | 13.0 (3.54) | 7.36 |
| Cundinamarca | 3.25 (8.07) T = 142,975 | 2'794,656.2 [2'543,338–3'085,522] | 37.4 (15.4) | 13.38 |
| Guainía | nd | 46,370 [43,291–49,473] | 0.40 (0.55) | 8.63 |
| Guajira | 1.61 (8.06) T = 72,608 | 855,974 [803,092–927,506] | 4.00 (4.00) | 4.67 |
| Guaviare | nd | 80,822.8 [77,328–84,716] | 1.60 (2.51) | 19.8 |
| Huila | 3.87 (9.43) T = 175,723 | 1'086,841 [1'061,405–1'111,844] | 26.0 (8.75) | 23.92 |
| Magdalena | 4.91 (13.2) T = 13,553 | 1'319,255.8 [1'268,980–1'388,832] | 9.20 (7.92) | 6.97 |
| Meta | nd | 1'021,131 [987,232–1'052,125] | 16.6 (12.7) | 16.26 |
| Nariño | nd | 1'621,210 [1'608,726–1'628,981] | 25.4 (16.8) | 15.67 |
| Norte de Santander | 4.45 (12.3) T = 96,860 | 1'467,692 [1'409,900–1'565,362] | 8.20 (6.91) | 5.59 |
| Putumayo | nd | 340,939.8 [327,856–353,759] | 5.20 (5.59) | 15.25 |
| Quindío | nd | 535,620 [526,484–547,855] | 13.0 (7.97) | 24.27 |
| Risaralda | nd | 936,713 [923,443–952,511] | 24.6 (10.9) | 26.26 |
| San Andrés | 4.05 (12.7) T = 10,677 | 61,567 [61,406–62,482] | 0.00 (0.00) | 0 |
| Santander | 5.39 (12.6) T = 175,976 | 2'157,188.6 [2'097,069–2'237,587] | 28.2 (8.84) | 13.07 |
| Sucre | 3.56 (11.7) T = 32,638 | 893,516.6 [867,701–928,984] | 14.8 (9.31) | 16.56 |

*(Continued)*

**Table 1.**  (Continued)

| Department | Precipitation[a] | Population[b] | OT cases[c] | OT cases per inhabitants |
|---|---|---|---|---|
| | Daily mean in mm (SD) T = Total precipitation in 5 years in mm. | Mean 5 years [range] | Annual mean (SD) | Average cases in 5 years per million inhabitants |
| Tolima | 4.95 (12.0) T = 167,612 | 1'327,187.4 [1'320,911–1'335,313] | 27.0 (10.9) | 20.34 |
| Valle del Cauca | nd | 4'445,393.2 [4'397,194–4'506,768] | 109 (50.6) | 24.52 |
| Vaupés | nd | 39,940.8 [37,638–42,721] | nd | nd |
| Vichada | nd | 105,304.2 [100,392–110,599] | nd | nd |
| Not defined[a] | na | na | 54.6 (54.9) | nd |

nd: no data, na: not applicable

[a]Available data from the National Institute of Hydrology, Meteorology, and Environmental Studies (IDEAM) about daily average precipitation from January 01, 2015 to December 31, 2019.

[b]Data from the retroprojections of the National Administrative Department of Statistics.

[c]Mean of cases between 2015 and 2019 by department, which are calculated as follows: total of records of OT in the department from 2015 to 2019 divided in 5 years. Cases of OT in Colombia by departments between 2015 and 2019 were retrieved from the System of Information of Social Protection.

Health System has one of the most prominent coverages in Latin America, encompassing 50 million inhabitants, which represents 98.88% of the population in 2021 [18].

We searched demographic data of patients grouped using the RIPS of the SISPRO database to obtain the epidemiological descriptions of all the OT registers (coded as ICD-10: B58.0) [16]. We applied a filter by year (2015–2019) and type of diagnosis (i.e., new confirmed, repeated confirmed, or not confirmed) to delimit the results. We only used the "new confirmed" filter for every month of each year (2015–2019), assuming that they were the new cases diagnosed in that period by physicians following international standards. We confirmed that the resulting dynamic table only reported consultations with the unique identifier in a single column filter to ensure no repeated patients.

## Precipitation data

Additionally, we requested precipitation data for all Colombia's departments from January 1, 2015, to December 31, 2019, via corporate email to the National Institute of Hydrology, Meteorology, and Environmental Studies (IDEAM) [19], which has been gathering information on precipitation from 1,567 weather stations distributed in the natural regions of Colombia since 1981. They sent a total of 2,157 files, from which we selected 825 files. We excluded files with no information on daily precipitation for the years 2015 through 2019. Each file corresponded to a meteorological station with the hourly precipitation measurement data. We further averaged these hourly data into daily measurements and calculated the daily average. We obtained data from most departments of Colombia (Table 1).

## Statistical analysis

In this study, we compared the exposure (daily average precipitation) in the period when events (cases of OT) occurred with exposures in nearby referent periods to examine the differences in exposure that may contribute to the differences in the daily number of OT cases by departments. A time-stratified case-crossover design was adopted to regulate potential confounders (e.g., age and sex) using self-control and excluding the long-term impact of

precipitation by stratification of time by departments. We used the calendar month as the time stratum to control the effects of long-term trends, seasonality, and day of the week.

Some departments, such as Amazonas, Vaupés, and Vichada, were excluded from the analysis because they did not have information about precipitation and cases of OT. Moreover, other departments, including Arauca, Caquetá, Casanare, Guainía, Guaviare, Meta, Nariño, Putumayo, Quindío, Risaralda, and Valle del Cauca, were excluded due to the lack of data from the meteorological stations.

We performed a quasi-Poisson regression, controlling for the over-scattering problem, combined with a distributed lag non-linear model (DLNM) to estimate the precipitation's non-linear and lag influence on the appearance of OT in Colombia. The DLNM class is used to describe the associations where the dependency between an exposure and an outcome is delayed in time, it means, the lag. The DLNM is based on the definition of "cross-based," two-dimensional space of functions to reflect the non-linear exposure responses and delay structure of the association [20,21]. First, a base model with natural cubic splines was fitted to optimize control for confounding factors, and DLNM was built as follows:

$$Y_t \sim Poisson(\mu_t)$$

$$Log(\mu_t | disease_{ye}) = A\alpha + \beta\, precipitation_{i,t} + ns(time, df = 7) + day$$

A quasi-Poisson regression model combined with a time-stratified case-crossover design and DLNM was built as follow: day of observation (t); the count of OT cases on $t$ ($Y_t$); the disease count at time $t$ ($disease_{ye}$); $\alpha$ is an intercept; $precipitation_{i,t}$, as the $i$th precipitation concentration on $t$; and $ns(t)$ represents the cross-basis function with natural spline $df_{2i-1} = 7$ for precipitation.

Although the $Tg$ can survive until months in humid conditions, to capture the complete lag–response curve, the maximal lag of precipitation was set in our model to 14 days, for the sake of simplification and without loss of generalization; meanwhile, this maximal lag was assigned to the length of the case and control periods. In addition, a 3-day duration was specified to be the maximal lag of meteorological factors. The df and maximum lag days for precipitation determination referred to the Akaike information criterion for quasi-Poisson (Q-AIC), which could produce the relatively superior model.

To identify the influence of precipitation, we calculated for and presented the relative risk (RR). We calculated the single-day lag influence and cumulative lag influence (lag0–1, lag0–6, lag0–8, lag0–10, lag0–12, lag0–13, and lag0–14) to effectively depict the characteristics of the association between precipitation and OT cases.

We computed the incidence rates using a spatial generalized linear mixed model for unit area data, where the response variable is Poisson, for the precipitation map, fitted using log-linear regression (a log link and error distribution of Poisson) [22–24]. With the Poisson spatial model, the exponents of the β coefficients are equal to the incidence rate ratio (RR). The β coefficient was estimated using as covariate the exposure to precipitation. For this analysis, we used the data of the new cases of OT reported in 2019 and the precipitation data published by the IDEAM for 2019 [25]. Additionally, we performed a Pearson correlation test using the available data for 2019 in the departments evaluated previously by Gomez-Marin et al. [10] (Guajira, Atlántico, Norte de Santander, Santander, Caquetá, and Quindío) to evaluate if their results were reproducible for other types of $Tg$ infection, such as OT. A larger temporal subanalysis could not be done due to the lack of data in the previous years. All analyses were conducted using R version 3.5.1 with the package for fitting the DLNM and GNM package for conditional quasi-Poisson regression and CARBayes [26].

### Bias control

Selection bias may occur due to the large number of filters that can be applied to diagnostic data in SISPRO and the existence of ICD codes, such as H30.9 (unspecified chorioretinal inflammation), where some retinochoroidal toxoplasmosis could be misclassified, leading to underestimation. To control bias, we only included patients with a new diagnosis of OT ("new confirmed"). Additionally, it is essential to note that ophthalmologists most commonly use the B58.0 code, SISPRO-based studies using ICD-10 have shown 83.4% of concordance with the medical record, and ICD-10 has demonstrated acceptable accuracy in studies on uveitis using big data [27–29].

## Results

There were 1,741 new cases of OT in Colombia between the years 2015 and 2019. The departments with the highest annual average number of OT cases were Valle del Cauca (109; standard deviation [SD] 50.6), Bogotá (106; SD 38.5), and Antioquia (88.2; SD 19.0). More detailed information regarding the number of OT cases registered in each department is presented in Table 1. We found a significant difference in the distribution of OT cases among different departments, which implies that each department behaves differently (*P*< 0.01). Additionally, regarding the average precipitation in millimeters of water (mm), the departments with the highest daily average precipitation were Chocó (12.8 mm; SD 23.8), Caldas (9.78 mm; SD 19.5), and Antioquia (8.02 mm; SD 16.5). The summary statistics for precipitation of departments for which data were available are shown in Table 1 [19].

### Cumulative exposure–response curves

The cumulative exposure–response curves show the cumulative influence of precipitation as the independent variable and the probability of having OT expressed as RR as the dependent variable for each department. The data from most departments showed a downslope cumulative exposure–response curve (Fig 1). Due to the size of the information presented and the variability of the data, we decided to present the results of our analysis of the seven departments that represent the most characteristic patterns in Fig 2. The departments of Guajira and Bogotá showed a directly proportional relationship between precipitation and greater risk of new OT development. In the department of Cauca, although the increase in precipitation is not directly proportional to the risk of new OT, it is accompanied by an increase in risk that becomes constant over time. Regarding Chocó, although a protective effect is evident in lag0–6 days, from lag7–14 days, there is evidence of a directly proportional relationship between increased precipitation and risk of new OT.

Contrary to the abovementioned data, the characteristic pattern of the other regions showed an inversely proportional relationship between precipitation and the risk of new OT, as in the case of Huila. Furthermore, in Sucre, precipitation increase the risk during the first 12 days, but then this effect decreases. Finally, San Andrés acted as a control chart, because no OT cases were recorded during this period, based on the Colombian Ministry of Health database. More information from all the departments is presented in S1 Fig.

### Single-day lag–response curves

The single-day lag–response curves showed daily precipitation as the independent variable and the probability of having OT, expressed as a RR, as the dependent variable, for each department (Fig 3). The curve for Bogotá demonstrated a variable effect at different lag days, reaching the highest RR on the lag10 day. Contrary to the Chocó curve, the Cauca and Sucre curves showed a decrease in RR; consequently, as time progresses, exposure loses its effect on

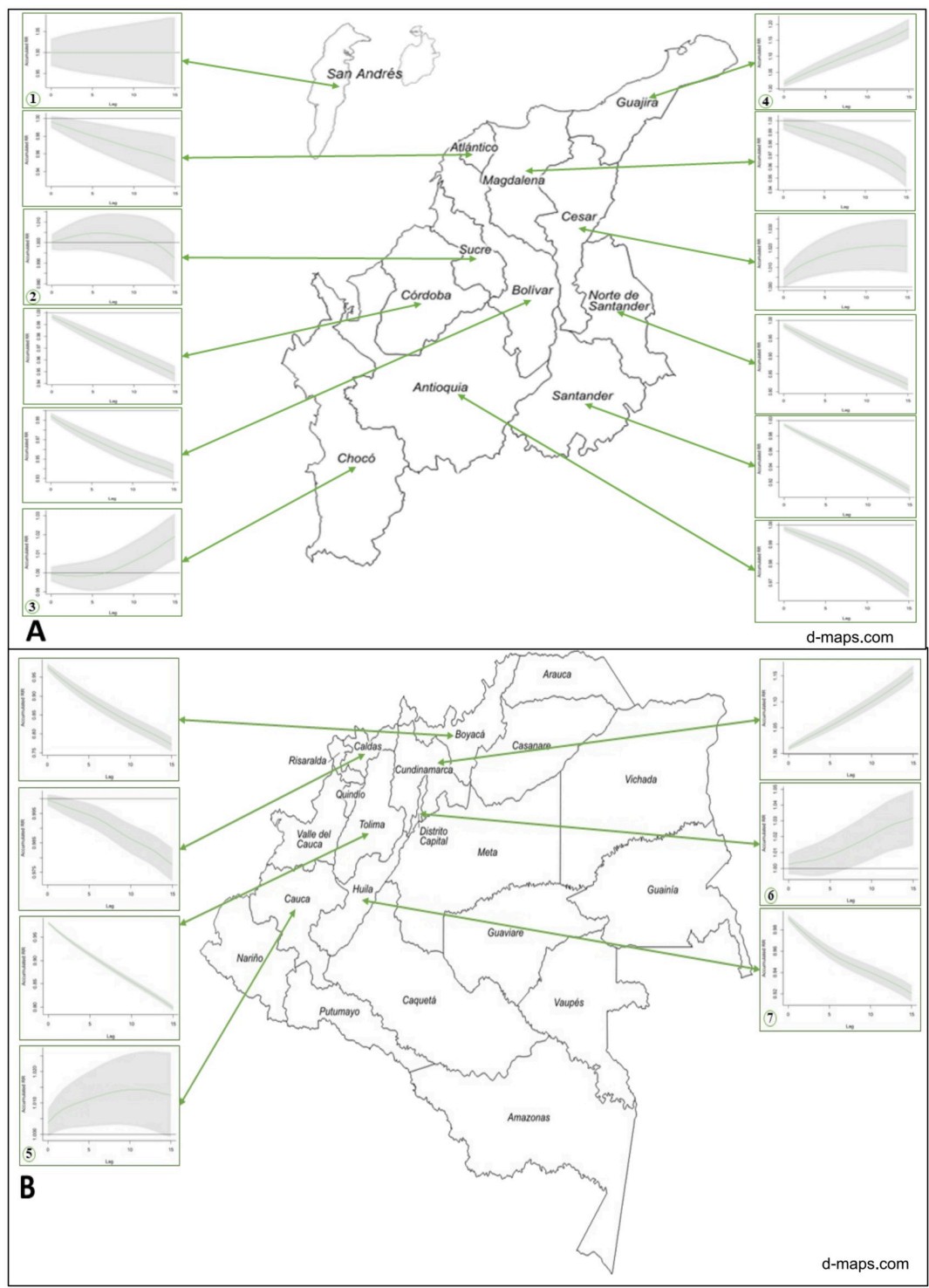

**Fig 1. Cumulative exposure–response curves for the association between precipitation and the new cases of ocular toxoplasmosis (OT) and its distribution in Colombia.** A. Evidence of the effect of precipitation in the northern departments. B. Effect of precipitation in the southern departments. Due to the climatic diversity secondary to the country's geography, it is not possible to completely segment the northern and southern regions of the country in any of the Köppen–Geiger classification system [15]. *The seven departments of interest are (1) San Andrés, (2) Sucre, (3) Chocó, (4) Guajira, (5) Cauca, (6) Bogotá, and (7) Huila. Map is from https://d-maps.com/carte.php?num_car=4095&lang=es.

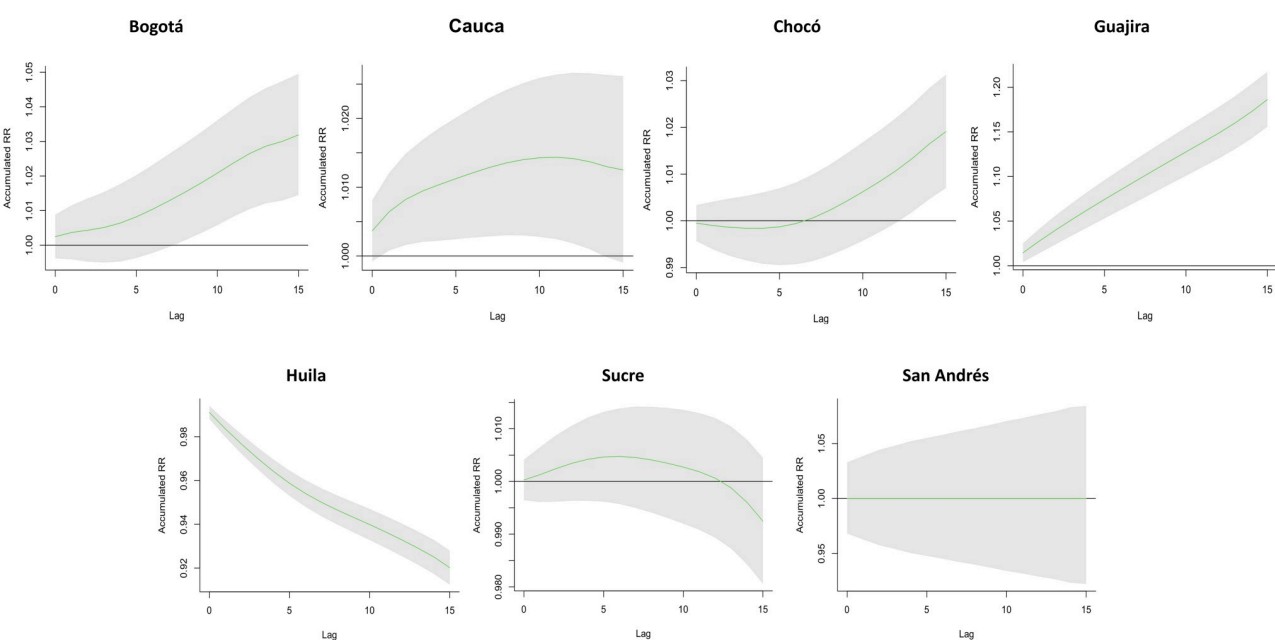

**Fig 2. Cumulative exposure–response curves of precipitation on ocular toxoplasmosis (OT) new cases at lag0–14, using a DLNM from 2015 to 2019 in seven selected departments of Colombia.** The solid green line represents the RR. Shaded areas represent 95% confidence interval (CI). Each lag refers to a period of the day (morning and afternoon); therefore, two lag equals 1 day. The cumulative exposure–response curves have a J-shape for Chocó and Bogotá, indicating that when a certain amount of precipitation accumulates, the RR increases. Similarly, the curve shows a growing trend for Guajira, meaning that the accumulative RR of OT increases proportionally with the precipitation exposure. In contrast, Huila's curve decreases indicating that the accumulated risk decreases while exposure increases (precipitation). The curve for Cauca shows an inverse U-shape with a transient increase in the RR, but the curve for Sucre has a similar pattern without a clear association because it crosses the zone of no effect. Finally, the graph for the department of San Andrés serves as a control case because it has no reported cases and maintains a constant null risk.

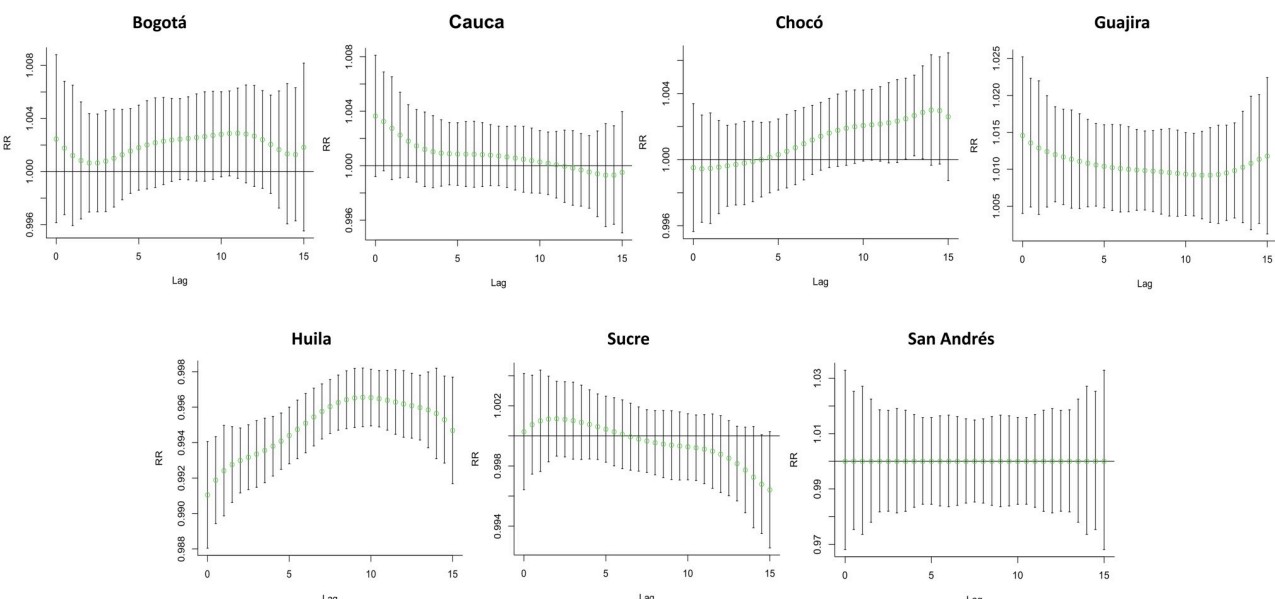

**Fig 3. Single-day lag–response curves on ocular toxoplasmosis for precipitation at a lag0–15 model in seven departments of Colombia from 2015 to 2019.** This figure represents the effect of a single-day precipitation for 15 consecutive days, among the total cases, using the DLNM model in seven departments of Colombia from 2015 to 2019.

**Table 2. Influence of precipitation in the risk of ocular toxoplasmosis on different lag days.**

| Department | lag 0–6<br>Mean of relative risk RR (95% CI) | lag 6–10<br>Mean of relative risk RR (95% CI) | lag 10–12<br>Mean of relative risk RR (95% CI) | lag 13–15<br>Mean of relative risk RR (95% CI) |
|---|---|---|---|---|
| Bogotá | 1.001(0.998–1) | 1.001(0.999–1) | 1.001(1–1) | 1.001(0.999–1) |
| Cauca | 1.002(1–1) | 1(0.999–1) | 1(0.999–1) | 0.9998(0.998–1) |
| Chocó | 0.9998(0.998–1) | 1(0.999–1) | 1.001(1–1) | 1.001(1–1) |
| Guajira | 1.007(1–1.01) | 1.005(1–1.01) | 1.005(1–1.01) | 1.005(1–1.01) |
| Huila | 0.9955(0.994–0.997) | 0.9975(0.997–0.998) | 0.9983(0.997–0.999) | 0.998(0.997–0.999) |
| San Andrés | 1(0.984–1.02) | 1(0.992–1.01) | 1(0.992–1.01) | 1(0.991–1.01) |
| Sucre | 1(0.998–1) | 1(0.999–1) | 0.9996(0.999–1) | 0.9991(0.998–1) |

the risk of developing OT, crossing the line of no effect at lag12 and lag7 days, respectively. Furthermore, Guajira showed a decreasing curve that remains over the no effect line, indicating precipitation as a constant risk factor for OT. Contrarily, Huila showed an increasing curve and a peak of RR near the lag10 day that always remains under the line of no effect as a protective factor. The San Andrés' graph showed a continuous or straight line, working as a control. Information on the other departments is available in S2 Fig.

Table 2 summarizes the cumulative influence of precipitation on different lag days for all new cases of OT in Bogotá, Cauca, Chocó, Guajira, Huila, San Andrés, and Sucre. The results showed that high precipitation could significantly increase the risk of OT cases, and the accumulated influence increased proportionally in some departments. The maximum RR value was 1.005 (95% CI 1–1.01), appearing in lag13–15 days in Guajira. Contrarily, in Chocó, the incidence rate seems to be the lowest, but it can be seen in the temporal analysis that the risk increases when the exposure increases (Table 2). Information regarding RR in other departments is available in S1 Table.

In the analysis by the mean precipitation in 2019 (Fig 4A), Chocó and Antioquia and the departments bordering the Andean mountains have a higher precipitation frequency than the others. However, this does not correspond with the departments with the highest risk of OT in the same period, as the highest incidence rates were found in Caquetá and Caldas (Fig 4B). Nevertheless, since the average of precipitation tends to vary over time, the risk could also change. This analysis supported the use of DLNM to observe the dynamic effect of exposure over time. Additionally, we estimated a positive β coefficient (0.03) with $P < 0.05$ between precipitation and OT risk overall, which suggests that precipitation significantly influences the response variable (risk of OT) in all the evaluated departments. The analysis for other years could not be carried out since we did not have their daily data.

## Validation of a previous hypothesis

We performed a Pearson correlation test for the available data to compare with the results of Gómez-Marin et al.[10] on the correlation between precipitation and *Tg* infection rates in some Colombian departments. Considering that IDEAM did not provide sufficient raw data from Caquetá and Quindío, we collected the average precipitation data for these departments from the secondary data available for 2019 [25]. This analysis showed a correlation between precipitation and incidence of OT of 0.84 ($P < 0.05$), supporting the results of their hypothesis. Fig 5A and 5B demonstrates the assumption.

## Discussion

Some studies relate rainfall with congenital toxoplasmosis development, OT relapses, and outbreaks. Regarding congenital infection, Gómez-Marin's et al. study [10] found a statistically

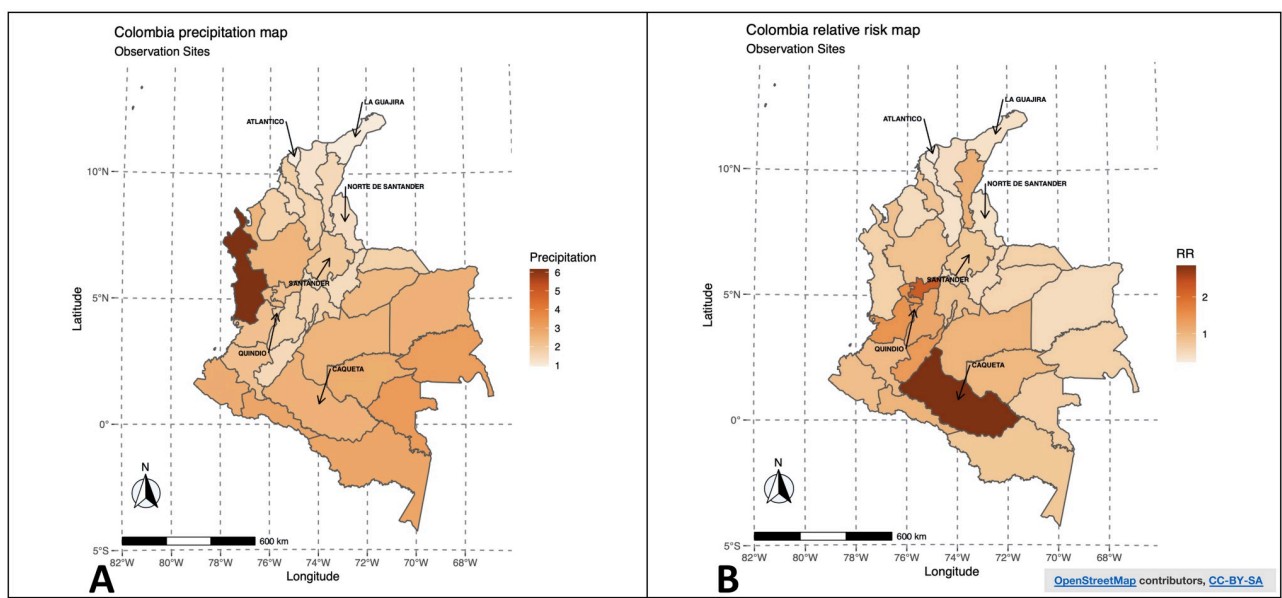

**Fig 4. Precipitation in Colombia and incidence rate of ocular toxoplasmosis (OT).** (A) The precipitation map of the departments in Colombia, which shows the average annual value of precipitation in each department for the year 2019; we were able to collect the information for all the departments from the National Institute of Hydrology, Meteorology, and Environmental Studies (IDEAM)[25]. (B) The estimated relative risk of OT with the fitted conditional autoregressive (CAR) model for the data available in 2019. Caquetá, Caldas, and Quindío have a higher relative risk, and departments with less precipitation, such as Atlántico and La Guajira, have a lower relative risk. Map was created based on https://cran.r-project.org/web/packages/leaflet/index.html.

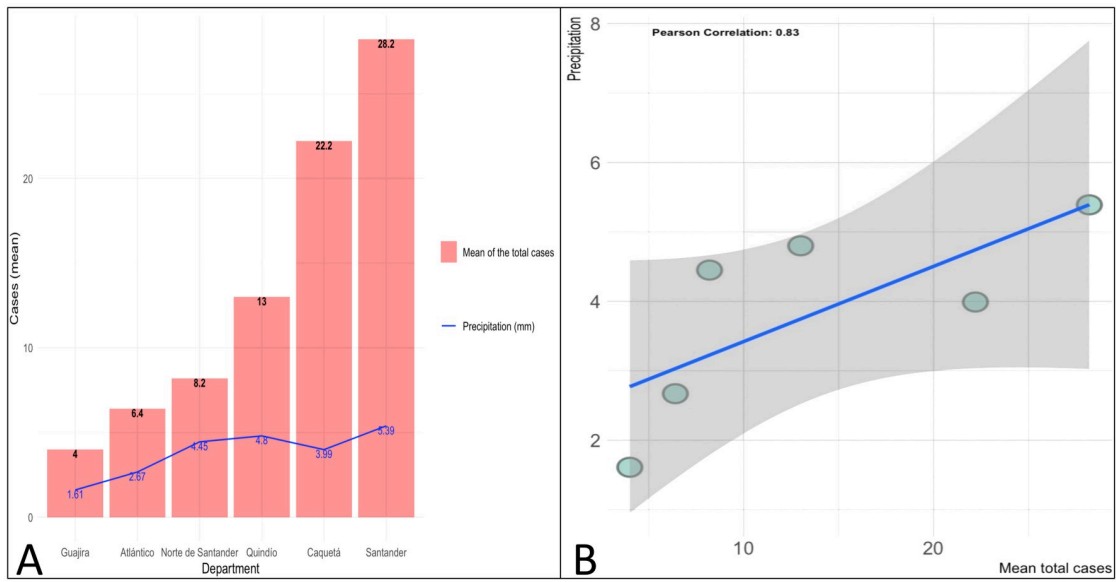

**Fig 5. The mean number of cases of ocular toxoplasmosis (OT) and precipitation by department in 2019.** (A) Represents the average number of cases of ocular toxoplasmosis (OT) in salmon bars and average precipitation by the department in the blue line (For Caquetá and Quindío only the information for 2019 was available). (B) The graph shows an incremental trend in the number of OT cases directly proportional to the precipitation in most cases (Pearson correlation test 0.84; *P* < 0.05). The data for this subanalysis were taken from IDEAM [25].

significant association between annual rainfall and congenital toxoplasmosis. With respect to OT relapses, a study carried out in Argentina found that the frequency of toxoplasmic reactivation episodes increases when precipitation increases and that the mean annual rainfall could be a predictor of the frequency of reactivations; for every mm of rainfall, there was a 2% increase in the reactivation episodes (OR = 1.002, 95% CI = 1.000–1.003, $P$ = 0.019) [11].

As in other foodborne infectious diseases, the incidence of OT is influenced by environmental factors. For example, OT outbreaks have been associated with contaminated water and undercooked food [30–35]. Regarding outbreaks and rainfall, a study from British Columbia found that the increased precipitation between 1994 and 1995 was associated with a waterborne outbreak of toxoplasmosis. They noted an increase in the number of OT cases, either congenital or acquired [34]. In Coimbatore, India, in September 2004, 178 patients had de novo retinochoroidal lesions and positive antibodies for *Tg*. All cases were closely related to the Siruvani reservoir, which supplies water to the city of Coimbatore and received rainwater due to increased rainfall that year, which could be related to the development of new OT cases [35]. Nevertheless, these studies did not find a statistical correlation between rainfall and OT.

Our results and those of the abovementioned studies could be directly related to the viability of *Tg* oocysts in soil. In one study, after 100 days, 7.4% and 43.7% of oocysts survived under dry and damp conditions, respectively [12]. *Tg* oocysts survive up to 54 months in cold water [13]. Furthermore, water contaminated with the parasite can potentially cause large toxoplasmosis outbreaks [13]. Small parasites can be filtered out by coagulation, flocculation, and settling used in most municipal water treatment systems in developed countries. However, developing countries do not always have these complex systems; therefore, oocysts approximately 8–12 μm in size could not be filtered in the water [13,36,37].

In the current study, we conducted a temporary analysis, which has never done before on the subject, evidencing a clear benefit to establishing the impact of different precipitation levels over time and revealing findings that cannot be appreciated in the clinical setting immediately after rainfall episodes. This type of analysis gives a clearer perspective of the effect of a climatic event on the population. We found a significant difference in the number of cases per department that correlates with what was found in the patterns of single-day lag–response curves and cumulative exposure–response curves. We can state that rainfall in Colombia is a factor that influences the presentation of OT (as a risk or protective factor), probably depending on the population behavior since, in some cases, unknown variables significantly reduce this risk, as shown in S1 and S2 Figs.

We suspect that the orographic barrier that surrounds the Andes mountains and induces the formation of local and regional high complexity climates in Colombia affects precipitation behavior, turning it highly variable [38]. Furthermore, the influence of thermodynamic processes in the Atlantic and Pacific oceans through the El Niño, La Niña-Southern Oscillation cycle can influence the climate and precipitation differently in each region across the country [39]. Both phenomena explain the variability of precipitation in our data by departments.

The cumulative exposure–response curve showed a downward trend for most of the departments with available data (Fig 1), indicating that the influence of precipitation on OT decreased as the exposure accumulated. However, the single-day lag–response curves were highly variable between departments, with no clear predominant trend (shown in S2 Fig). The cumulative exposure–response curves and Table 2 show that Chocó, Bogotá, Cesar, Cauca, and Guajira have a pattern that is contrary to that observed in the rest of the country, with an increase in the RR for OT directly proportional to the mean precipitation. This supports the theory that precipitation is a risk factor. Chocó is one of the areas with the highest rainfall in the world, with an average annual precipitation of 8,000 to 13,000 mm. The data demonstrated

that precipitation in this department increased the risk of OT from lag0–6 (RR 0.99) to lag13–15 days (RR: 1.01) [14].

It is also important to consider how cultural habits could explain the variability in RR of OT in the proposed model. In Chocó, when the rainy season begins, the RR may decrease because the population does not go out to collect water or work in the rivers. Previous studies have shown that behaviors tend to vary significantly in populations with a higher risk of floods and rising rivers [40,41]. However, over time the risk tends to increase in Chocó. It may also be due to the presence of contaminated waters, as Chocó is one of the country's poorest regions with the most limited access to health, food, and aqueduct; in fact, only 35% of the population have access to this essential service [42–44].

As for Bogotá, a direct relationship between precipitation and OT risk was observed. This finding provides strong data, as this is the capital city with better health access and reporting systems. Therefore, the general decreasing risk in most departments in Colombia, one of the endemic countries, may be strongly affected by the underreporting of the disease throughout the country, suggesting that the risk may be even greater than expected, for example, inadequate reporting technique could explain that San Andrés does not have new cases reported [45,46].

Our results suggest that many other environmental variables could influence the relationship between OT and precipitation, and they must be assessed locally and through a temporal analysis. For example, the wind is an ecological variable involved in oocyst sporulation and dispersal (contamination) in water sources, soil, and food [47–49]. Studies have even proposed that oocysts could aerosolize. An epidemiological study showed an outbreak in people from an equine stable, suspecting that the cause was aerosolized oocysts. However, their attempts to isolate the parasite in different samples did not show the presence of oocysts at any time [48]. Interestingly, in the Guajira, the average wind speed values are higher than nine m/s. In the Andean region, a wind corridor reaches average wind speed values above five m/s [50,51]. Therefore, this and other environmental factors should be analyzed in further studies to confirm an association. We propose it as an example of variables that confound the effect of precipitation on the risk of OT.

## Limitations

Given that our study was a population-based one, we did not have access to the clinical records of patients and could not confirmed the serological status. However, we only included new confirmed cases with code (B 58.0), which ophthalmologists commonly use. Additionally, data from the national databases may have inherent coverage and content errors that underestimate the number of new cases. Moreover, most of the ophthalmology and tertiary care centers are located in capital cities, such as Bogotá, which can generate some bias in the analysis. Furthermore, this explains why departments with a high incidence of OT, such as Quindío, did not have too many cases. Finally, due to limitations in the data provided by IDEAM, some departments could not be evaluated in detail and therefore were not included in the analysis. From the 2,157 files that they have sent for the specified period, only 825 (38%) files had the required information. For instance, in the data obtained from the National Registry in 2017, the department of Chocó had 30 municipalities, of which only 8 (26.7%) reported information on the monitoring of the quality of water for human consumption [52].

## Conclusions

In the present study, our temporal analysis showed the impact of different precipitation levels over time on the RR for OT. It revealed findings we could not otherwise appreciate in the

clinical setting immediately after rainfall episodes. There were variable tendencies according to the department evaluated. Depending on the region, the precipitation increased, decreased, or showed no relationship with RR for OT. We hypothesize that other sociodemographic and environmental variables, such as wind, could influence the RR. Additionally, the correlation analysis shows a directly proportional relationship between the increased number of cases and the increase in precipitation. Data regarding precipitation should be analyzed individually for each region and particular context. The influence of other environmental, behavioral, and sociodemographic factors should be examined in the departments with a higher risk.

## Supporting information

**S1 Fig. Cumulative-response curves for the association between precipitation and new cases of ocular toxoplasmosis in Colombia, 2015–2019.**
(DOCX)

**S2 Fig. Single-day lag-response curves for the association between precipitation and new cases of ocular toxoplasmosis in Colombia, 2015–2019.**
(DOCX)

**S1 Table. Mean of Relative risk (RR) and 95% confidence intervals (95% CI) in each department of Colombia of ocular toxoplasmosis cases.**
(DOCX)

## Acknowledgments

We thank Enago for the manuscript language edition.

## Author Contributions

**Conceptualization:** Laura Boada-Robayo, Danna Lesley Cruz-Reyes, Carlos Cifuentes-González, Alejandra de-la-Torre.

**Data curation:** Laura Boada-Robayo, Danna Lesley Cruz-Reyes, Ángela Paola Vargas-Largo.

**Formal analysis:** Danna Lesley Cruz-Reyes.

**Methodology:** Laura Boada-Robayo, Danna Lesley Cruz-Reyes, Carlos Cifuentes-González, William Rojas-Carabali.

**Project administration:** William Rojas-Carabali, Ángela Paola Vargas-Largo.

**Resources:** William Rojas-Carabali, Ángela Paola Vargas-Largo.

**Software:** Danna Lesley Cruz-Reyes.

**Supervision:** Carlos Cifuentes-González, William Rojas-Carabali, Alejandra de-la-Torre.

**Validation:** Carlos Cifuentes-González, William Rojas-Carabali, Alejandra de-la-Torre.

**Visualization:** Alejandra de-la-Torre.

**Writing – original draft:** Laura Boada-Robayo, Carlos Cifuentes-González, William Rojas-Carabali, Alejandra de-la-Torre.

**Writing – review & editing:** Laura Boada-Robayo, Danna Lesley Cruz-Reyes, Carlos Cifuentes-González, William Rojas-Carabali, Ángela Paola Vargas-Largo, Alejandra de-la-Torre.

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
