## [Decision Letter · Decision Letter 0]

21 May 2022

Dear Dr de-la-Torre,

Thank you very much for submitting your manuscript "Exploring the association between precipitation and population cases of ocular toxoplasmosis in Colombia" for consideration at PLOS Neglected Tropical Diseases. As with all papers reviewed by the journal, your manuscript was reviewed by members of the editorial board and by several independent reviewers. In light of the reviews (below this email), we would like to invite the resubmission of a significantly-revised version that takes into account the reviewers' comments. 

We cannot make any decision about publication until we have seen the revised manuscript and your response to the reviewers' comments. Your revised manuscript is also likely to be sent to reviewers for further evaluation.

Sincerely,

Christine A. Petersen

Deputy Editor

Christine Petersen

Deputy Editor

Reviewer's Responses to Questions

**Key Review Criteria Required for Acceptance?**

**Methods**

-Are the objectives of the study clearly articulated with a clear testable hypothesis stated?

-Is the study design appropriate to address the stated objectives?

-Is the population clearly described and appropriate for the hypothesis being tested?

-Is the sample size sufficient to ensure adequate power to address the hypothesis being tested?

-Were correct statistical analysis used to support conclusions?

-Are there concerns about ethical or regulatory requirements being met?

Reviewer #1: There is a clear objective for this study, but no state hypothesis from the authors. Additionally the authors need to make clear if there was any missing precipitation data and the potential impacts of missing precipitation data. the authors also need to account for population differences in the geographical regions of interest presented in this study.

Reviewer #2: (No Response)

**Results**

-Does the analysis presented match the analysis plan?

-Are the results clearly and completely presented?

-Are the figures (Tables, Images) of sufficient quality for clarity?

Reviewer #1: the results followed and analyses discussed in the methods section. Figure 1 is not clearly presented. Supplemental material 1 and 2 should be included as in-text tables for this manuscript.

Reviewer #2: (No Response)

**Conclusions**

-Are the conclusions supported by the data presented?

-Are the limitations of analysis clearly described?

-Do the authors discuss how these data can be helpful to advance our understanding of the topic under study?

-Is public health relevance addressed?

Reviewer #1: the authors conclusions are supported by the results. Additionally the authors do a nice job discussing the limitations of their work and data.

Reviewer #2: (No Response)

**Editorial and Data Presentation Modifications?**

Reviewer #1: (No Response)

Reviewer #2: (No Response)

**Summary and General Comments**

Reviewer #1: This study evaluated the relationship between precipitation and ocular toxoplasmosis incidence. I have some comments below on your manuscript:

a. Introduction: The authors should consider adding more information on the protozoan lifecycle and biology that allows it survive in conditions outside of its hosts. Additionally the authors should discuss routes of infection and what can influence infection. 

b. Methods: the authors should note what months and years they gathered data for in regards to their outcome data (population data section) in a similar way it is mentioned in the precipitation data. 

c. Methods: The authors state “825 files” were selected on line 148. Do these files represent data from 825 unique stations for the four years they were interested in researching? Or is it 825 unique files with data from all weather stations available?

d. Methods: the authors should state if there was missing data in regards to precipitation data, and if there was missing data how was in handled?

e. Methods: the authors should make remarks on how they accounted for population size differences between geographic regions that could account for some differences they see in counts of OT per region. 

f. Results: Supplementary Material 1 and 2 should be included as in-text tables as they describe the authors data used for their results

g. Results: In line 209, the authors describe different departments. I believe departments is referring to geographical regions of Columbia. The authors should make this more clear throughout the manuscript and provide greater discussion in their methods that these are analyses are performed for each geographic region. 

h. Results: the authors should give a greater description of what a characteristic pattern is for line 217. 

i. Results: The legend in figure 4B would indicate that precipitation is associated with a decrease in risk since RR < 1. Is the legend presented in this figure accurate?

j. Results: in line 273-275 the authors should state there is an association between precipitation and increased OT rather than the causal language currently used. Direct causality is difficult to prove and assess in a population study. 

k. Results: For data presented in supplementary Material 2, is this average of daily averages, or the average of hourly precipitation recorded from weather stations? Authors should make this clearer. 

l. Results: Figure 1 is difficult to read the graphs. Either splitting the map into northern and southern compartment to make a two part figure would be helpful. Or placing an asterisk (or some denoting feature) on the 7 characteristic patterns would provide improved readability of these graphs. Or decreasing the size of the map and increasing the graphs. 

m. Results / Discussion: Authors should discuss why in some regions there is large variability in RR (Cauca, Sucre, San Andres, etc.) where in others there is very little RR variability

Reviewer #2: In this manuscript, the authors describe the association between precipitation and occurrence of ocular toxoplasmosis (OT) cases in different location in Colombia. OT is a serious clinical problem in South America, and studies to understand the interaction between condition and its environment are more than welcome to enable efficient public health measures on a local level. As previous studies (cited in the manuscript) showed, precipitation is one of the important factors. This kind of study is well adapted for publication in PLOS Neglected Tropical Diseases. The senior author has a solid publication list in OT clinical and basic science. The authors collected a huge data set to explain different ways of how precipitation affects OT cases, more than enough to justify an article. Unfortunately, the manuscript is written in a language and style apparently destined to meteorology experts, including a lot of formulas, techniques and interpretations specific to this area. In my opinion, an article in PLOS NTD should be accessible to readers trained in infectiology and tropical health science. Being such a reader, I considerably struggled to understand the methods, results and conclusions of the study. So, to be able to evaluate the manuscript correctly, I suggest the following points:

1 Clearly explain the methods used, and the terms employed for interpretation (lag time…). For example, the authors state that the filter … was applied for each year (l. 140), but the term ‘year’ never appeared in the results. In l. 154, it should be explained what exactly means ‘case period’ and ‘nearby referent periods’. The two following sentences are, as generally all this statistical analysis (= methods sensu stricto), very difficult to understand for the average infection/tropical health reader. The second paragraph of this section seems to be important, but should be clearly explained. This does, of course, not exclude some technical formulas, when well explained.

2 The results are also difficult to decipher. The main findings should be explained in clear statements for each figure. The legend to Fig.3 is very difficult to understand, to give just one example.

3 The text treating Fig4 and Table 1 (l.270-277) is quite difficult to follow, for different reasons: The methods employed are not clearly explained to the average reader (cf. point 1), the language of some sentences lack themselves clarity, e.g. l.272-273, and finally, the principal findings of the Table and Fig. are not clearly resumed in the text.

4 The discussion also lacks clear interpretation of data, stating basically that rainfall can both be risky and protective, for unknown ‘population behavior’ (l. 311f) reasons. Apart from the wind, at the end of the discussion, none is really detailed, and maybe taken as example for specific districts.

In summary, I think that the manuscript contains valuable information, but should be explained in a more commonly understandable fashion and language. I have just some more general points:

5 Most of the cited articles relate high precipitation levels and toxoplasmosis outbreaks to failures of the filtration system in public water supply. Here, it should be at least discussed if more rain acts through longer oocyst survival or more oocysts leaking into the public water supply.

6 I admit that I did not entirely understand the meaning of the term ‘delay’ throughout the manuscript, but if this means that OT cases were observed with a delay of a few days after rainfall episodes, this has to be explained. Ocular affection usually takes time to develop following infection. Moreover, can new detected OT cases serve as measure for new general infections?

7 Do the data allow to confirm the results in Ref 9 (Gomez-Marin et al.) of a correlation between rainfall and T. gondii infection rates in Colombian districts?

PLOS authors have the option to publish the peer review history of their article (what does this mean?). If published, this will include your full peer review and any attached files.

Reviewer #1: No

Reviewer #2: No
---

## [Decision Letter · Decision Letter 1]

15 Aug 2022

Dear Dr de-la-Torre,

We are pleased to inform you that your manuscript 'Exploring the association between precipitation and population cases of ocular toxoplasmosis in Colombia' has been provisionally accepted for publication in PLOS Neglected Tropical Diseases.

Best regards,

Christine A. Petersen

Section Editor

Christine Petersen

Section Editor

Reviewer's Responses to Questions

**Key Review Criteria Required for Acceptance?**

**Methods**

-Are the objectives of the study clearly articulated with a clear testable hypothesis stated?

-Is the study design appropriate to address the stated objectives?

-Is the population clearly described and appropriate for the hypothesis being tested?

-Is the sample size sufficient to ensure adequate power to address the hypothesis being tested?

-Were correct statistical analysis used to support conclusions?

-Are there concerns about ethical or regulatory requirements being met?

Reviewer #1: The objective of the study is clearly articulate with a testable hypothesis. the study design, data, and analysis are clearly stated. The population is clear and is appropriate for the hypothesis. Sample size is sufficient for adequate power.

Reviewer #2: (No Response)

**Results**

-Does the analysis presented match the analysis plan?

-Are the results clearly and completely presented?

-Are the figures (Tables, Images) of sufficient quality for clarity?

Reviewer #1: The analysis presented matches the analysis plan described in the methods. The results are clear and completely presented and show a vast improvement over the previous draft. In particular figures and figure legends are more clear.

Reviewer #2: (No Response)

**Conclusions**

-Are the conclusions supported by the data presented?

-Are the limitations of analysis clearly described?

-Do the authors discuss how these data can be helpful to advance our understanding of the topic under study?

-Is public health relevance addressed?

Reviewer #1: Conclusions are supported by the data presented and limitations are clearly presented.

Reviewer #2: (No Response)

**Editorial and Data Presentation Modifications?**

Reviewer #1: (No Response)

Reviewer #2: (No Response)

**Summary and General Comments**

Reviewer #1: Large improvement over previous submission. Methods and results are more clear and reader friendly.

Reviewer #2: According to the reviewers’ remarks, the manuscript has been ameliorated. I have just still a few remarks:

1 When I mentioned ‘language’, it did not mean the English proficiency, but the technical language used throughout, which is actually quite difficult to understand for an average expert in infectious diseases. So, every finding and the implication for oocyst infectivity should be explained in usual epidemiologic terms, in addition to the already present technical terms, of course. This should also include some statements why the effect is seen so fast after a single episode of rainfall, when the authors state that oocysts are viable for several months in humid conditions, as found in Colombia.

2 Fig.5: It is good that this analysis has been included, but it seems to me that the type of graph is not adapted to a correlation analysis. Furthermore, ‘mean of total cases’ and ‘precipitation’ should be specified.

3 The number of OT cases should also be calculated per inhabitants, as the departments vary considerably in population density.

4 Just one remark on (English) language, l. 419: ‘Lousy reporting technique’ should be substituted by a more neutral term.

PLOS authors have the option to publish the peer review history of their article (what does this mean?). If published, this will include your full peer review and any attached files.

Reviewer #1: No

Reviewer #2: No

---

## [Editor Report · Acceptance letter]

15 Sep 2022

Dear Dr de-la-Torre,

We are delighted to inform you that your manuscript, "Exploring the association between precipitation and population cases of ocular toxoplasmosis in Colombia," has been formally accepted for publication in PLOS Neglected Tropical Diseases.

Best regards,

Shaden Kamhawi

co-Editor-in-Chief

Paul Brindley

co-Editor-in-Chief
